# Species Differences in Ezetimibe Glucuronidation

**DOI:** 10.3390/metabo14110569

**Published:** 2024-10-22

**Authors:** Shalom Emmanuel, Eric A. Asare, Ting Du, Huan Xie, Dong Liang, Song Gao

**Affiliations:** Department of Pharmaceutical Science, College of Pharmacy and Health Sciences, Texas Southern University, 3100 Cleburne Street, Houston, TX 77004, USA; s.emmanuel2414@student.tsu.edu (S.E.); e.asare2673@student.tsu.edu (E.A.A.); du.ting@tsu.edu (T.D.); huan.xie@tsu.edu (H.X.); dong.liang@tsu.edu (D.L.)

**Keywords:** ezetimibe, glucuronidation, species differences, UGT

## Abstract

Background: Peclinical and clinical studies have revealed that ezetimibe, an approved cholesterol-absorption inhibitor, is rapidly and extensively metabolized to a more potent metabolite, ezetimibe glucuronide. Since different species are commonly used in the pharmacokinetic and pharmacodynamic studies of ezetimibe, it is essential to determine the species difference in glucuronidation of ezetimibe in order to accurately evaluate ezetimibe’s pharmacokinetics and pharmacodynamics. The purpose of the study was to compare species differences in ezetimibe glucuronidation rates using intestinal microsomes from humans, rats, mice, monkeys, and dogs. Method: Intestinal microsomes from different species were used to assess the ezetimibe glucuronidation rates. Multiple substrate concentrations at 0.5, 2, 5, 10, 20, 30, 40, and 50 µM were tested and fitted into the Michaelis–Menten model to determine the enzyme kinetic parameters. Results: The results showed that the glucuronidation rates with these tested species were significantly different. Kinetic studies revealed that the maximum metabolic rate (V_max_) was higher in monkeys (3.87 ± 0.22 nmol/mg/min) than that in rat (2.40 ± 0.148 nmol/mg/min) and mouse (2.23 ± 0.10 nmol/mg/min), and then human (1.90 ± 0.08 nmol/mg/min) and dog (1.19 ± 0.06 nmol/mg/min). The CLint was an 8.17-fold difference among these species, following the order of mouse > dog > human > rat = monkey. Conclusions: The study revealed that the rate of ezetimibe glucuronidation in the intestine was different in different species and has an impact on ezetimibe glucuronidation in the intestine. When analyzing the pharmacodynamics, pharmacokinetics, or toxicology of ezetimibe using different models, these species differences must be taken into consideration.

## 1. Introduction

Ezetimibe, a potent cholesterol-absorption inhibitor, has gained prominence as a crucial component in the management of hypercholesterolemia [1]. It is known to function as an inhibitor of cholesterol absorption by preventing the small intestine from absorbing cholesterol. Combining ezetimibe with statins can lower the LDL-C level and reduce the risk of cardiovascular events. The FDA has authorized using ezetimibe and atorvastatin together to lower LDL-C in individuals with primary or mixed hyperlipidemia [2]. Ezetimibe has been demonstrated to lower inflammatory markers and atherosclerosis, with a satisfied safety and tolerability profile. Besides reducing cholesterol levels, ezetimibe has demonstrated the ability to enhance insulin sensitivity and diminish insulin resistance linked to nonalcoholic fatty liver disease [3]. All things considered, ezetimibe is a safe, effective, well-researched, and commonly used medication for hypercholesterolemia [4].

The glucuronidation of ezetimibe is mediated mainly by UDP-glucuronosyltransferase (UGT). Pharmaceutical studies have revealed that ezetimibe is rapidly and extensively converted into the metabolite ezetimibe glucuronide (Eze-G), mainly mediated by UGTs, specifically UGT1A1 in the liver and intestine, after being administered orally. In vitro studies showed that incubation of ezetimibe with human liver microsomes resulted in the formation of Eze-G. Additionally, ezetimibe was metabolized into glucuronide rapidly when incubated with intestinal microsomes. Ex vivo studies showed that more than 92% of ezetimibe was present in the glucuronide form in the portal vein after oral administration [5], suggesting that the intestine is the major metabolic organ for orally administered ezetimibe, even though the liver is usually considered a major metabolic organ for most of the drugs administered through oral route. PK studies have shown that the majority of the ezetimibe (80–90%) is presented in the glucuronide form in the plasma, and the oral bioavailability of ezetimibe is just 35–65% due to extensive metabolism in the intestine [6]. Additionally, ezetimibe undergoes enterohepatic recycling mediated by glucuronide, resulting in a long half-life of 22 h in the plasma [7].

Studies have indicated that the Eze-G exhibits greater potency than its parent form. Both ezetimibe and its glucuronide reduce plasma cholesterol when taken orally; however, the glucuronide is more effective [8]. Research has shown that the in vivo activity in many species is correlated with the rank order of affinity of ezetimibe glucuronide for its molecular target, Niemann–Pick C1 Like-1 (NPC1L1), with the glucuronide exhibiting greater affinity and potency [9]. The glucuronide form’s higher binding affinity for ezetimibe’s molecular target, NPC1L1, has been ascribed to its improved efficacy. Because of this, ezetimibe’s glucuronide form is thought to be more powerful than its parent form, which helps explain its effectiveness at lowering plasma cholesterol levels [10].

Various animal models, including mice, rats, dogs, and monkeys, have been utilized in pharmacokinetic and pharmacodynamic studies of ezetimibe. Additionally, veterinarians sometimes prescribe ezetimibe for dogs to treat hyperadrenocorticism [11]. Given that the glucuronide metabolite produced in the intestine is more potent than ezetimibe itself [8], it is crucial to investigate whether and how the glucuronidation of ezetimibe varies across different species, because differences in glucuronidation activity in the intestine across species may potentially affect dosing translation to humans. Furthermore, species differences in ezetimibe glucuronidation could have significant implications for conclusions drawn from drug–drug interaction studies, where these animal models are frequently used [12]. Drug–drug interactions, a critical aspect of therapeutic management, may be influenced by the interplay between ezetimibe and co-administered drugs, each subject to unique metabolic pathways [13]. Understanding the species-specific variations in glucuronidation can aid in predicting potential interactions and guide dosage adjustments to optimize therapeutic outcomes [14]. The purpose of this study is to demonstrate the species difference in ezetimibe disposition in the intestine, which is the major drug target organ, in different species, as these species are commonly used in preclinical studies.

## 2. Materials and Methods

### 2.1. Chemicals and Materials

Ezetimibe (98% purity) and ezetimibe β-D-glucuronide (98% purity) were purchased from Cayman Chemical (Ann Arbor, MI, USA). Alamethicin and uridine diphosphoglucuronic acid were obtained from Sigma-Aldrich (St. Louis, MO, USA). Methanol and acetonitrile (HPLC grade) were obtained from VWR International (Radnor, PA, USA). Saccharolactone, rutin, formic acid, and magnesium chloride were purchased from Sigma-Aldrich (St. Louis, MO, USA). All other chemicals and solvents were of the highest grade commercially available. The intestinal microsomes from pooled male Cyno monkeys (10 mg/mL), pooled male beagle dogs (10 mg/mL), pooled male CDI mice (10 mg/mL), pooled male IGS rats (10 mg/mL), and pooled human (10 mg/mL) were purchased from Sekisui Xenotech (Kansas City, KS, USA).

### 2.2. Method

*Ultra-performance liquid chromatographic (UPLC) method*. An ezetimibe stock solution (20 mM) was prepared in dimethyl sulfoxide (DMSO). The stock solution was diluted to 1 mM with 50% methanol and further diluted to 100 µM working solution. A standard curve was generated by diluting the working solution to concentrations ranging from 100 μM to 0.78 μM in KPI buffer, which contained 0.6% formic acid in acetonitrile and 5 mM of the internal standard (Rutin). The internal standard (IS, 5 mM Rutin) was prepared in 50% methanol. The varying concentrations of the working solution were then vortexed and centrifuged at 14,000 rpm for 15 min. Aliquots (100 uL) of the centrifuged samples were then loaded into the sample tubes to be analyzed using Waters Acquity UPLC. The chromatographic conditions were mobile phase—0.1% formic acid in water (Solvent A) and acetonitrile (Solvent B); stationary phase (Column)—C18 1.7 µm, 2.1 × 50 mm; temperature, 45 °C; flow rate, 0.45 mL/min; injection volume, 10 µL; run time, 5 min; wavelength, 232 nm; and gradient elution method, 5% B for 0.5 min, followed by a linear increase to 95% B at 3 min, 3–3.5 min at 95%, 3.5–4.5 min at 95%, and a re-equilibrium at 5% B at 4.5 min.

*In vitro glucuronidation reaction using intestinal microsomes.* An in vitro Phase II glucuronidation reaction was carried out using a published protocol that involved the use of uridine diphosphate glucuronic acid (UDPGA) as cofactor, saccharolactone as beta-glucuronidase inhibitor to drive the forward reaction to release the metabolite, potassium phosphate buffer (KPI) to mimic physiological conditions, and alamethicin as a surfactant for the reaction [15]. Based on our preliminary studies, an appropriate protein concentration (0.01 mg/mL) was used to incubate the different intestinal microsomes for 1 h at 37 °C, with different substrate concentrations (0.5, 2, 5, 10, 20, 30, 40, and 50 µM). At a protein concentration of 0.01 mg/mL, the rate of product formation was linear with respect to time (1 h) and saturating with respect to substrate and cofactor concentration. All the samples and solutions were brought from the freezer and put on ice. The components were pipetted into the Eppendorf tubes in the following order: substrate, KPI, Solution B (alamethicin, saccharolactone, and magnesium chloride), Solution A (uridine diphosphate glucuronic acid and potassium chloride), and enzyme to make a final volume of 170 µL. The solution was incubated at 37 °C. Triplicate samples (100 µL) were taken after 1 h, and the reaction was terminated with 50 µL of solution containing 0.6% formic acid and 5 mM internal standard in methanol at each time point and vortexed for 1 min. The samples were then centrifuged at 14,000 rpm for 15 min. The supernatant (100 µL) was then collected and loaded into the UPLC for analysis. The results were analyzed by determining the ezetimibe/IS peak area ratios and plotting them against their respective concentrations obtained from a standard curve. They were then analyzed using GraphPad Prism (Version 10.3) software.

*Enzyme kinetic study with ezetimibe*. The enzyme kinetic study was carried out by assaying 0.01 mg/mL intestinal microsomes from the different species, which were incubated at 37 °C with varying concentrations of the substrate (0.5 µM, 2 µM, 5 µM, 10 µM, 20 µM, 30 µM, 40 µM, or 50 µM, respectively) for 1 h.

*Data analysis*. A standard curve was generated from the results of the serial dilution with both the parent and the glucuronide compounds by determining peak area ratios of absorbance and plotting them against the concentration of the analytes using Microsoft Excel. UPLC was used to quantify the metabolites formed, and GraphPad was used to generate the Michaelis–Menten plot to determine enzyme activity and kinetics (CLint, km, and V_max_). *t*-test and one-way ANOVAs were used to compare the means of triplicate samples analyzed and assess the differences between intestinal microsomes.

## 3. Results

### 3.1. Ezetimibe Glucuronidation

When ezetimibe was incubated with intestinal microsomes, an additional peak was observed in the UPLC analysis. The retention time of this additional peak was identical to that of standard Eze-G. Additionally, the UV spectrum of this additional peak showed two absorption peaks at 227 and 232.1 nm, which were similar to those of the standard Eze-G. Thus, this additional peak was identified as Eze-G. Figure 1 shows the conversion of ezetimibe to its glucuronide form, Eze-G (Figure 1A). The parent compound elutes at a retention time of 2.9 min, Eze-G elutes at 2.5 min, and the IS (Rutin) elutes at 1.9 min (Figure 1B,C).

### 3.2. Ezetimibe Glucuronidation Activities in Intestinal Microsomes of Different Species at 5 μM

Ezetimibe glucuronidation activities in the intestinal microsomes of humans, mice, rats, dogs, and monkeys were initially compared at a substrate concentration of 5 μM. (Figure 2). The results showed that the metabolic rate of dog intestinal microsomes was the slowest, at 0.72 ± 0.01 nmol/min/mg. In contrast, the highest metabolic rate was significantly different (2.8-fold higher), with mouse intestinal microsomes at 2.02 ± 0.03 nmol/min/mg. Additionally, the results indicated that the metabolic rates of monkey, mouse, and dog intestinal microsomes were comparable to that of human intestinal microsomes.

### 3.3. Kinetics for Ezetimibe Glucuronidation by Intestinal Microsomes of Different Species

The V_max_ and km were calculated and are summarized in Table 1. Table 1 shows the significant difference across all species studied. The km values in mice and dogs were comparable with each other, whereas other groups showed statistical difference from one another. Meanwhile, in regard to the V_max_ values, there was a significant difference across all species analyzed. The intrinsic clearance (CLint) was also calculated, and the values represented show the ability of the species to be able to clear out the drug after metabolism.

#### 3.3.1. Kinetics of Ezetimibe Glucuronidation in Human Intestine Microsomes

Ezetimibe glucuronide formation fits into the Michaelis–Menten model in the human intestine microsome (Figure 3A). The V_max_ values in the human intestine were recorded to be 1.90 ± 0.08 nmol/mg/min, and it displayed a significant difference compared to the other species investigated. The km value in the human intestine was calculated to be 1.33 ± 0.36 µM. The intrinsic clearance (V_max_/km) value for glucuronide was estimated to be 1.43 ± 0.01 µL/min/mg (Table 1).

#### 3.3.2. Kinetics of Ezetimibe Glucuronidation in Rat Intestine Microsomes

Ezetimibe glucuronide formation fits into the Michaelis–Menten model in the rat intestine microsome (Figure 3B). The V_max_ value in the rat intestine microsome was 2.40 ± 0.14 nmol/mg/min, which was significantly different from that in mouse and dog intestine microsomes, but significantly lower than that in monkey intestine microsomes. The km value in the rat intestine was calculated to be 4.10 ± 1.03 µM, which was significantly lower than that in monkey intestine microsome. The intrinsic clearance (V_max_/km) value for glucuronide was estimated to be 0.58 ± 0.01 µL/min/mg (Table 1).

#### 3.3.3. Kinetics of Ezetimibe Glucuronidation in Male Cyno Monkey Intestine Microsomes

Ezetimibe glucuronide formation was shown to fit into the Michaelis–Menten model in the monkey intestine microsome (Figure 3C). The V_max_ value in the monkey intestine microsome was 3.87 ± 0.22 nmol/mg/min, which was significantly different from the other species studied. The km value in the male Cyno monkey intestine was calculated to be 8.01 ± 1.60 µM, which is significantly higher than the km of other species of microsome. The intrinsic clearance (V_max_/km) value for glucuronide was estimated to be 0.47 ± 0.02 µL/min/mg (Table 1).

#### 3.3.4. Kinetics of Ezetimibe Glucuronidation in Male CDI Mouse Intestinal Microsomes

Ezetimibe glucuronide formation fits into the Michaelis–Menten model in the mouse intestine microsome (Figure 3D). The V_max_ value in a male mouse intestine microsome was recorded to be 2.23 ± 0.10 nmol/mg/min, which was higher than that in the dog intestine microsome, but lower than those in the other species studied. The km value in the male CDI mouse intestine was calculated to be 0.58 ± 0.19 µM, similar to the km of beagle dog intestine microsome. The intrinsic clearance (V_max_/km) value for glucuronide was estimated to be 3.84 ± 0.01 µL/min/mg (Table 1).

#### 3.3.5. Kinetics of Ezetimibe Glucuronidation in Beagle Dog Intestine Microsomes

Ezetimibe glucuronide formation fits into the Michaelis–Menten model in the beagle dog intestine microsome (Figure 3E). The V_max_ value in a male beagle dog intestine microsome was 1.19 ± 0.06 nmol/mg/min, displaying a significant difference compared to the other species. The km value in the male beagle dog intestine was calculated to be 2.59 ± 0.64 µM. The intrinsic clearance (V_max_/km) value for glucuronide was estimated to be 2.17 ± 0.01 µL/min/mg (Table 1).

## 4. Discussion

Despite the therapeutic significance of ezetimibe glucuronide, which is known to be more potent than ezetimibe itself [16], the specific species differences in the glucuronidation process remain largely unknown. This knowledge gap is particularly critical, as it directly impacts the efficacy, drug–drug interactions, and/or safety of ezetimibe studies using models from different species, such as PK/PB or PK/PD modeling. This study found significant interspecies variability of ezetimibe glucuronidation using intestinal microsomes of mice, rats, dogs, monkeys, and humans. The reason why mouse and rat microsomes were tested is that these two species are the commonly used animal models in ezetimibe pharmacokinetics and pharmacodynamic studies. Dog microsomes were tested because ezetimibe was also used to lower total and low-density lipoprotein cholesterol in hypercholesterolemic dogs with hyperadrenocorticism by veterinarians, even though this drug is not approved in dogs [11]. Monkey microsomes were tested because their metabolic pathways closely resemble those of humans.

We first determined the metabolic rates using a substrate concentration at 5 μM, and the results showed species difference (Figure 2). Then, we determined the V_max_ and km using multiple substrate concentrations (Figure 3). The metabolic rate allowed us to know the reaction rate at a particular substrate concentration, and the V_max_ allowed us to find out the maximum capability of the enzyme functions so that we can comprehensively evaluate the species difference.

Among these tested species, monkeys had the highest V_max_, followed by mice and rats at similar rates, whereas dogs had the lowest V_max_ among the tested species (3.25-fold). Mouse intestine microsomes showed the highest intrinsic clearance compared to other species. These differences highlight the need to consider species-specific metabolic characteristics when extending experimental results to human physiology, especially in the context of drug metabolism and pharmacokinetics. Such insights are critical for optimizing preclinical investigations and improving the translation of experimental outcomes to clinical settings.

The formation of ezetimibe glucuronide in the human intestine is mainly catalyzed by UGT1A4 and UGT1A3 [16]. UGT polymorphisms have been widely studied in drug metabolism, with results indicating that certain mutations can significantly affect glucuronidation activity across individuals. In this study, the microsomes were prepared from a pooled sample of genders and individuals, sourced from a reputable vendor. We do not expect polymorphisms to play a role in the differences observed across species in this study. Additionally, some UGT isoforms in animals differ from those in humans, and recombinant animal UGT isoforms are currently not commercially available, making it impossible to investigate the contribution of UGT isoforms to species differences in the glucuronidation of ezetimibe.

The V_max_ value of 2.40 nmol/mg/min recorded for the glucuronidation of ezetimibe in the rat intestine microsome is higher than the V_max_ value of 1.90 nmol/mg/min recorded for the glucuronidation of ezetimibe in the human intestine microsome. This difference in V_max_ values in comparison to the human intestine microsome could be due to species-specific differences in the expression and activity of the UDP-glucuronosyltransferase (UGT) enzymes responsible for the glucuronidation of ezetimibe [17]. The UGT enzymes are known to exhibit interspecies variability in their expression levels, substrate specificity, and kinetic properties [5]. Therefore, the higher V_max_ value observed in the rat intestine microsome could be attributed to the higher expression or activity of the UGT enzymes responsible for ezetimibe glucuronidation in rats compared to humans [17]. The V_max_ values recorded in different studies can be influenced by various experimental conditions, such as the source and quality of the microsomes, the assay conditions, and the specific UGT isoforms present in the samples, which may differ between the human, rat, monkey, beagle dog, and mouse studies [18].

The liver is commonly considered the major metabolism organ, and previous in vitro studies showed that liver microsomes can convert ezetimibe into Eze-G rapidly [15]. However, ex vivo studies using animal models showed that >90% of orally administered ezetimibe is glucuronidated in the intestine before the parent can reach the liver, as evidenced by the glucuronide/parent presenting in the portal vein [5]. We previously found that ezetimibe was extensively metabolized in the intestine into Eze-G, which was efficiently taken up by the hepatocytes via OATP1B1 and 1B2 transporters and then efficiently (91.4%) secreted into the bile to participate in recycling after being hydrolyzed by gut microflora [16]. Therefore, the intestine is believed to be where most of the ezetimibe dosed through the oral route is converted into glucuronide in the intestine, and the intestine is the major metabolic organ in vivo, even though liver microsomes can metabolize ezetimibe rapidly. Therefore, we determined the metabolic difference using the intestinal microsomes from different species in this study.

We first compared the metabolism using a single substrate concentration at 5 μM. The result showed that the metabolic rate with dog intestinal microsomes was significantly slower than that of humans, while the metabolic rate with monkeys was significantly higher than that of humans. Kinetic studies showed that all the kinetics with all the tested species fit the Michaelis–Menten model, which suggested that this process follows a saturable enzymatic kinetics mechanism. The CLint of mice was the highest among the tested species, while that of monkeys was the lowest (8.17-fold). Compared to humans, the CLint of rats and monkeys was significantly lower, while that of the dog and mouse was significantly higher. Intrinsic clearance (CLint) measures the intestine’s capacity to remove a medication without relying on blood flow or protein binding. A higher number indicates a quicker removal of the drug. These differences in comparison to the humans could be because of the species-specific differences in the expression and activity of the UDP-glucuronosyltransferase (UGT) enzymes responsible for the glucuronidation of ezetimibe [18]. It was reported that UGT1A1 and UGT1A3 are the major isoforms catalyzing ezetimibe glucuronidation using recombination human UGT isoforms [5]. The UGT enzymes are known to exhibit interspecies variability in their expression levels, substrate specificity, and kinetic properties [18].

The study focusing solely on species differences in the glucuronidation of ezetimibe while investigating only using microsomes mediated in vitro reaction presents some limitations. Comparing kinetic parameters (i.e., V_max_, km, and CLint) obtained from a microsome study between species can be difficult due to fundamental physiological variances between species. V_max_ is the highest rate of a metabolic reaction under ideal conditions, and it varies depending on body size, metabolic rate, enzyme kinetics, and organ function. These parameters vary greatly between species due to evolutionary divergence and adaptation to different ecological niches. For example, larger animals may have higher absolute V_max_ due to increased metabolic demands, yet smaller animals may have higher V_max_ relative to their body size. In vivo studies using different species are expected to validate the abovementioned findings.

Microsomes are derived from the endoplasmic reticulum of liver cells and contain a high concentration of UGTs, the enzymes that catalyze ezetimibe’s glucuronidation reaction. This is a traditional model in drug metabolism study, which can provide rapid insight into species difference. However, the data generated in a microsomes study usually require validations using in vivo studies, as the conditions differ significantly. For example, there is no blood flow in a microsome study. The results of this study provide a primary evaluation of ezetimibe’s glucuronidation by various species. More in vivo studies are suggested based on the results afforded in this study.

We did not test the UGT isoforms because the purpose of this study is to compare the overall ezetimibe metabolic difference in the intestine across species. Recombinant human UGT isoforms are usually used to determine the metabolic mechanism and drug–drug interaction. Additionally, recombinant animal UGT isoforms are currently not commercially available.

## 5. Conclusions

In conclusion, the findings of this study highlight variations in ezetimibe glucuronidation across species, particularly in the intestine. Therefore, it is essential to thoughtfully consider the distinctions in intestinal glucuronidation levels between primates and rodents during preclinical studies. This study also suggests that species differences have an impact on ezetimibe glucuronidation in the intestine, and when analyzing the pharmacodynamics and pharmacokinetics of ezetimibe, these species differences must be taken into consideration. Going forward, it is important to accent that these species are commonly used in preclinical studies and the study can be further looked into for further and future research.

## Figures and Tables

**Figure 1 metabolites-14-00569-f001:**
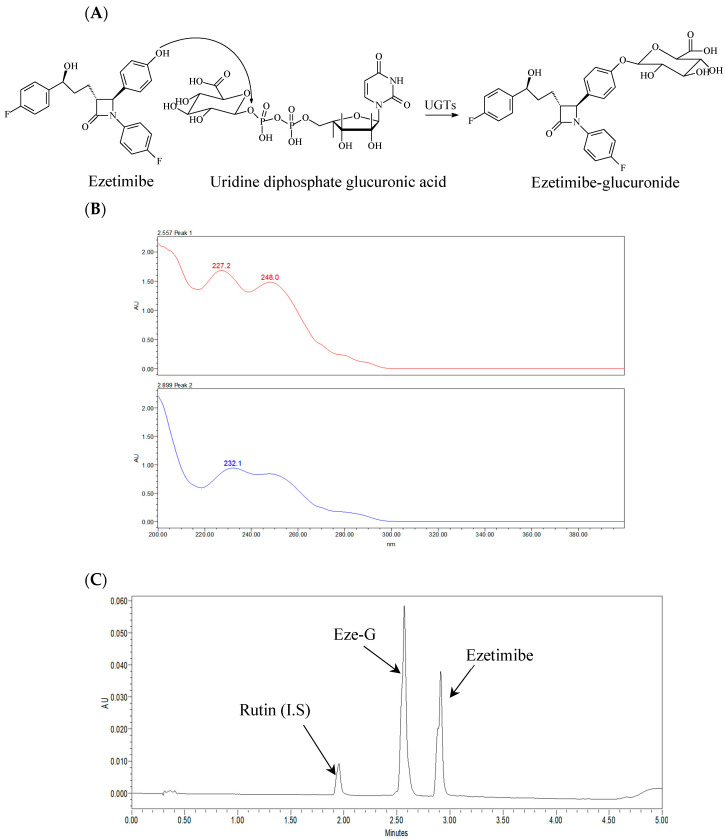
Chemical structures, a representative UPLC chromatogram, and UV spectra of ezetimibe and ezetimibe glucuronide (Eze-G). (**A**), Chemical structures of ezetimibe and its glucuronide and glucuronidation reaction; (**B**), UV spectra of the additional peak and ezetimibe-glucuronide; (**C**), A Representative Chromatogram of the analytes with I.S.

**Figure 2 metabolites-14-00569-f002:**
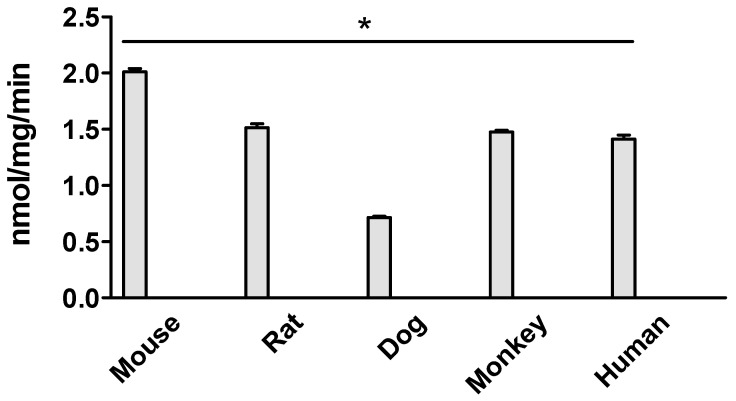
Ezetimibe glucuronidation in intestinal microsomes of mouse, rat, dog, monkey, and human. The substrate concentration is 5 μM, and the incubation is 1 h. The results show that the metabolic rate of dog intestinal microsomes is significantly lower than that of the other species (* *p* < 0.05, one-way ANOVA).

**Figure 3 metabolites-14-00569-f003:**
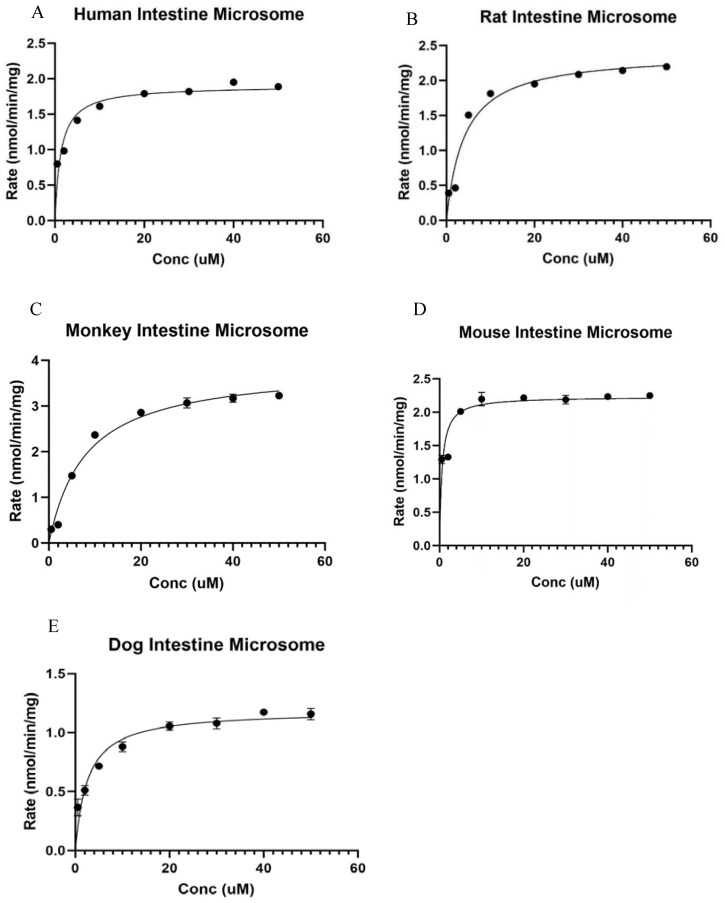
Kinetic study of Ezetimibe glucuronidation in intestinal microsomes of various species: (**A**) human, (**B**) rat, (**C**) monkey, (**D**) mouse, and (**E**) dog. The results showed that the V_max_ of rat live microsomes is significantly lower than the others.

**Table 1 metabolites-14-00569-t001:** Kinetic parameters for ezetimibe glucuronidation by different intestine microsomes of humans, rats, monkeys, mouse, and dogs.

Species	K_m_ (µM)	V_max_ (nmol/mg/min)	CL_int_ (µL/min/mg)
Human	1.33 ± 0.36	1.90 ± 0.08	1.43 ± 0.01
Rat	4.10 ± 1.03 *	2.40 ± 0.14 *	0.58 ± 0.01 ^#^
Monkey	8.01 ± 1.60 *	3.87 ± 0.22 *	0.47 ± 0.02 ^#^
Mouse	0.58 ± 0.19 ^#^	2.23 ± 0.10 *	3.84 ± 0.01 *
Beagle dog	2.59 ± 0.64 *	1.19 ± 0.06 ^#^	2.17 ± 0.01 *

Each value represents the mean ± SD of triplicates experiments. * Higher than that of a human (*p* < 0.05, *t*-test). ^#^ Lower than that of a human (*p* < 0.05, *t*-test).

## Data Availability

All the data are stored in the Research Infrastructure Core (RIC) under the Center for Biomedical and Minority Health Research (CBMHR) at TSU. The data are not publicly available due to university policies. However, all data are available for research purposes upon request.

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
