# Peer review of "Species Differences in Ezetimibe Glucuronidation"

_metabolites, 2024, doi:10.3390/metabo14110569_

Round 1
Reviewer 1 Report (Previous Reviewer 1)
Comments and Suggestions for Authors
Dear Authors,
The article “Species Differences in Ezetimibe Glucuronidation” introduces a new knowledge about Ezetimibe pharmacokinetics. This drug is commonly used medication for hypercholesterolemia in humans and could be effective also in the veterinary practice.
The article is well-structured, the methods are adequative.
There are some comments.
1. The answer to reviewer â„–5 is not mentioned in the revised manuscript. “The purpose of this study is to demonstrate the species difference in ezetimibe disposition in the intestine, which is the major drug target organ, in different species, as these species are commonly used in preclinical studies”.
Please, add this sentence or the part of it. It is important to accent that “these species are commonly used in preclinical studies” . The differences found can be taken into account in a future preclinical studies of the new drugs.
2. It is desirable to add some comments about the differences in “metabolic rate” and Vmax
3. It is desirable to discuss the justification of the usage of isolated intestinal microsomes as a pharmacokinetic model. What are the evidences in the literature?

Author Response
Dear Authors,
The article “Species Differences in Ezetimibe Glucuronidation” introduces a new knowledge about Ezetimibe pharmacokinetics. This drug is commonly used medication for hypercholesterolemia in humans and could be effective also in the veterinary practice. The article is well-structured.
There are some comments.
- The answer to reviewer â„–5 is not mentioned in the revised manuscript. “The purpose of this study is to demonstrate the species difference in ezetimibe disposition in the intestine, which is the major drug target organ, in different species, as these species are commonly used in preclinical studies”. Please, add this sentence or the part of it. It is important to accent that “these species are commonly used in preclinical studies”. The differences found can be taken into account in a future preclinical study of the new drugs.
-
It is desirable to add some comments about the differences in “metabolic rate” and Vmax
-
It is desirable to discuss the justification of the usage of isolated intestinal microsomes as a pharmacokinetic model. What are the evidences in the literature.
We thank this reviewer's comments and suggestions. We have revised the manuscript accordingly. Major changes are in red in the text. Please see attached point-by-point response.

Reviewer 2 Report (Previous Reviewer 2)
Comments and Suggestions for Authors
Please enhance the resolution of Figure 1. Fig. 1a. Please show a complete reaction by showing all reactants. Please also mention in the manuscript the advantage of using microsomes instead of pure UGTs.
Author Response
Please enhance the resolution of Figure 1. Fig. 1a. Please show a complete reaction by showing all reactants. Please also mention in the manuscript the advantage of using microsomes instead of pure UGTs.
We agree with this reviewer's comments and have revised the manuscript accordingly. Major changes are in red in the text. Please find point-to-point response in the attached response

This manuscript is a resubmission of an earlier submission. The following is a list of the peer review reports and author responses from that submission.
Round 1
Reviewer 1 Report
Comments and Suggestions for Authors
Dear Authors,
The article “Species Differences in Ezetimibe Glucuronidation” describes the rates of ezetimbe glucuronidation in intestinal microsomes of mouse, rat, dog, monkey, and human. The data obtained indicate the slower metabolism in dog microsomes, it could be important detail for the veterinary use of this drug.
There are some omissions. It is desirable to include the following information:
1. It is desirable to accent the final aim of the study in a more concrete way
2. The choice of the species should be explained
3. The possible role of UGT1A1 polymorphisms in the drug action. This issue have not been discussed in the paper. Could it be that the lower rate of glucuronidation was due to gene polymorphism?
4. It is not clear how the microsomes were prepared or they were commercially available and standardized? It should be mentioned: origin, viability, storage etc.
5. Is Ezetimibe used in the veterinary practice, particularly, for dogs and cats?
6. Is it possible to extrapolate the results of this study for the future clinical usage of Ezetimibe? What is the possible practical contribution of this study?
7. Does the concentration of Ezetimibe in this in vitro study (1-50 µM) correspond to it`s in vivo concentration in pharmacokinetic and pharmacodynamics studies?

Author Response
We agree with the reviewer's comments, which we used to improve our manuscritp. Please see the point-to-point response in the attached file. The major revised parts are in blue in the revision.

Reviewer 2 Report
Comments and Suggestions for Authors
Line 25. Please mention the name of substrate.
Line 13-15: "An Ultra-Performance Liquid Chromatography (UPLC) with UV detection between 227 to 250 nm, was used to quantify the ezetimibe and its glucuronide". Please remove from the abstract.
This study is very simple, which just determine the Vmax of UDP-glucuronosyltransferase (UGT) by using intestinal microsomes. The results are obtained through in-vitro experimentation and it would have been better if in-vivo experiments were performed to verify the results and check their applicability to the living species.
In the methodology, the method for preparation of intestinal microsomes is not given.
Please propose the mechanism how UDP-glucuronosyltransferase (UGT) perform the glucuronidation of ezetimibe.
Please add a flow diagram presenting the workflow.
The novelty of the manuscript is quite limited.
Author Response
Please find our response attached

Reviewer 3 Report
Comments and Suggestions for Authors
In the last decade, a series of papers have appeared in which the authors select the molecular structure of ezetimibe glucuronide and consider what effect it has in the intestine, and when analyzing the pharmacokinetics in dynamics and pharmacokinetics of ezetimibe, taking into account the species differences between individuals. These are large and significant reviews that reflect high importance for research, but at the same time, in my opinion, this work is presented extremely poorly for publication. The authors need to demonstrate the completeness and depth of the presented research in the illustrative component. Work out the reaction patterns and conduct a screening with a detailed description of each image. I recommend documenting any convincing reason. Unfortunately, this was not provided for a specific system. This manuscript contains a representative UPLC chromatogram and UV spectra of ezetimibe and ezetimibe glucuronide (Eze-G). The aim of the study was to compare species differences in the rate of formation of ezetimibe glucuronide using intestinal microsomes of humans, rats, mice, monkeys and dogs – where were these data selected from, why are these criteria informative for assessing the rate of glucuronidation of ezetimibe? It is difficult to assess the evidence base using only one liquid chromatography (UPLC) method with UV detection in the range from 227 to 250 nm. Kinetic studies have shown that the maximum metabolic rate has been found for several species of individuals. Why speed? The title of the work, conclusions, and literature leave much to be desired.
Therefore, this work requires serious improvements and cannot be recommended for publication.
Author Response
Please see our response attached.
